

# A structural equation model for the patient safety competency of clinical nurses

Jung-hyun Choi[1] and KyoungEun Kim[2]

[1] Department of Nursing, Namseoul University, Chonan, Chuncheongnam-do, Republic of South Korea
[2] Department of Child Welafre, Namseoul University, Cheoan City, Chungcheongnam-do, Republic of South Korea

## ABSTRACT

**Background:** Nurses are crucial for enhancing patient safety due to their continuous presence at patients' bedsides and close interactions with families and other healthcare providers. This study aims to examine the relationships among safety education, perception of patient safety culture, safety control, and patient safety competence in clinical nurses, while also exploring the mediating effect of perceptions on patient safety culture and safety control.

**Methods:** The study involved 165 nurses, including 10 males (6.1%) and 155 females (93.9%). Structural equation modeling (SEM) was used to test the hypothesized model, and data were analyzed using SPSS and AMOS programs.

**Results:** Significant positive correlations were among the frequency of attending safety education, the perception of patient safety culture, safety control, and patient safety competency. The number of safety education briefings attended did not directly influence patient safety competence; however, safety education for nurses indirectly influenced patient safety competence *via* the perception of patient safety culture and safety control. These findings suggest that enhancing safety education for nurses can improve patient safety competence by shaping their perceptions of patient safety culture and safety control.

## INTRODUCTION

Patient safety is a critical global issue (*Lee, Jang & Park, 2016*). Patient safety means eliminating preventable harm to patients during healthcare and reducing the risk in connection with healthcare to a minimum acceptable level (*Lee, Jang & Park, 2016*). The World Health Organization initiated the World Patient Safety Day 2020 Campaign to raise global awareness about the importance of patient safety (*World Health Organization, 2020*). In 2000, the Institute of Medicine (IOM) released "To Err is Human." It is reported that 251,454 people die from medical errors in US hospital per year and it was the third-leading cause of death in the USA (*Makary & Daniel, 2016*).

Patient safety competency means that all medical personnel have the knowledge, skills, and attitudes necessary to prevent medical errors and enhance patient safety (*Han, Kim & Seo, 2019*). In Korea, nurses are in a prominent position to ensure patients' safety because

Corresponding author
KyoungEun Kim, leejay48@nsu.ac.kr

they are the largest team in the health workforce. Therefore, the ability to improve patient safety should be incorporated into education for nurses (*Lee et al., 2014*), and education should be introduced early and continuously strengthened (*Wu & Busch, 2019*). Nurses who participated in patient safety training had higher levels of patient safety competency than those who did not (*Yan et al., 2021*). Patient safety competency implies the ethically responsible judgment of complex decision-making, such as prevention, recognition, reporting, and admitting safety problems, and it incorporates the communication aspect of safety and educating medical personnel about reporting and sharing committed mistakes. Therefore, enhanced critical thinking and teamwork training for patient safety are needed to improve patient safety competency among nurses (*Wu & Busch, 2019*).

The factors that influence patient safety competency can be distinguished as individual aspects such as safety education and safety control, and organizational aspects such as patient safety culture. To ensure patient safety, a patient safety culture has to be formed, which means that all professionals must believe that patient safety is the highest priority in patient care (*Nieva & Sorra, 2003*). Furthermore, a patient safety culture must consider all beliefs, values, attitudes, and behaviors shared by organizations, departments, and individuals in the hospital to prevent medical errors that may occur during the provision of medical services (*Halligan & Zecevic, 2011*). As nurses are optimally placed to discover mistakes and faults, safety culture is commonly considered a vital component of nursing (*Najjar et al., 2015*). Implementing patient safety into everyday nursing duties encourages frequent reporting and discussion of medical near misses, which will, in turn, improve patient care (*Amiri, Khademian & Nikandish, 2018*). To establish a patient safety culture in a health organization, some actions are needed to enhance the reporting of events, and non-punitive responses to errors need to be considered (*Amiri, Khademian & Nikandish, 2018*).

Safety control is related to deriving safe results in performing tasks. A high level of safety control reduces negative safety indicators and nurses with less safety control have less safety awareness (*Kim, 2019*). With a nurse who has a good level of safety control, patients will benefit both physically and psychologically (*Ganster & Fusilier, 1989*). Most nurses believe that they only need to report errors in the hospital ward, but they do not realize they need to report potential errors, as these can lead to broader issues in the future (*Cohen, 2001*). When facing immediate demands, nurses first must solve problems of conflicting priorities, which can cause harm in a variety of ways (*Tucker & Edmondson, 2003*). In these situations, nurses' safety control provides them with the correct judgment about actions that are putting patient safety at risk (*Kim, 2021*) and prevents nursing mistakes related to patient safety (*Kim et al., 2012*).

*Kim et al. (2012)* predicted that preventive education on nursing errors would improve safety control and contribute to safety competency. This is because the level of personal control over tasks has a direct impact on the nurse's perception of safety competency to assure the patient's well-being. Nurses are the frontline risk managers who care for patients 24 h a day. There have been many studies about patient safety competency (*Han, Kim & Seo, 2019*; *Beischel & Davis, 2014*; *Jin & Yi, 2019*), the perception

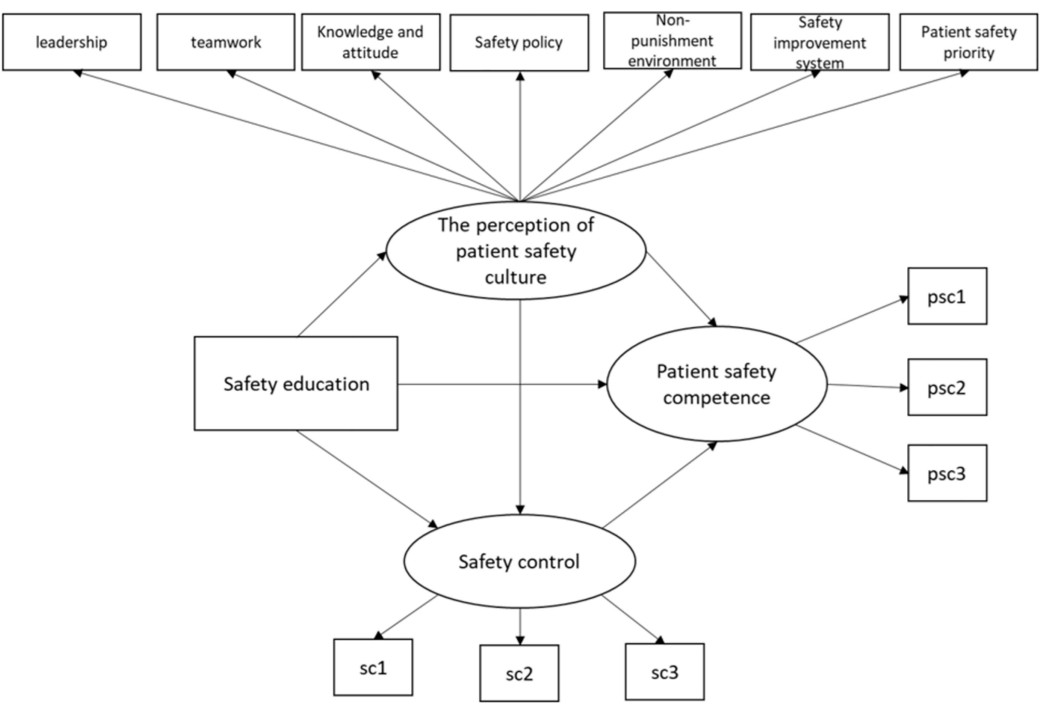

**Figure 1 Hypothesized model of the antecedents of patient safety competence.**

of safety culture (*Mardon et al., 2010*; *Wang et al., 2014*), and safety control (*Cohen, 2001*; *Tucker & Edmondson, 2003*). However, empirical evidence for the relationship between nurses' patient safety competency and perceived safety culture is limited (*Cohen, 2001*; *Tucker & Edmondson, 2003*). Patient safety competency research has purposefully sought to understand how variables are related within the individual or at the organization level, and how they interactively impact patient safety competency (*Ben-Tzion Karsh & Brown, 2010*).

Investigating and analyzing patient safety culture, patient safety competency, and the perception of safety control are necessary to prevent medical errors and improve patient safety.

Therefore, this study examines hospital nurses' perceptions of patient safety culture, patient safety competency, and levels of safety control to identify the structural relationships among factors that affect patient safety competency (see Fig. 1) and to provide essential evidence for patient safety competency interventions with nurses.

In the current study, the research questions are as follows:

• Research question 1: What is the relationship between nurses' safety education participation, the perception of patient safety culture, safety control, and patient safety competency?

• Research question 2: Is there a mediating effect of nurse's perception of patient safety culture and safety control on the relationship between patient safety education and patient safety competency?

## MATERIALS AND METHODS

### Design

This research is a descriptive correlation study to examine the variables affecting patient safety competence in clinical nurses.

### Sampling and data collection

Convenience sampling was conducted for nurses who have worked at H hospital and D hospital in Seoul, C hospital in Cheonan. The selection criteria included nurses who understood the purpose of the study and agreed to participate; had more than 1 year of clinical career.

The researcher visited the hospitals, explained the purpose of this study, and delivered a questionnaire to nurses who agreed to participate in this study. A questionnaire collection box was placed in the hospital nurse's office, and when the questionnaire was completed, it was put into the collection box. Data were received from nurses at three hospitals from October to December 2021.

The number of samples was calculated using the G Power 3.1.9.7 program. With an effect size of 0.15, a significance level of 0.05, and a power of 0.95, a total of 172 participants were calculated to be the required sample size for the multiple linear regression analysis, considering the predictor variables of the perceptions of patient safety culture, safety control, and safety education. Of the 191 copies retrieved, 165 (dropout rate: 13.6%) were used for the final analysis, excluding 24 with some skipped or double-checked for a single question.

### Participants

Table 1 shows the characteristics of the participants. Participants were 165 nurses: 10 males (6.1%) and 155 females (93.9%). Participants were from 20 to 58 years old (M = 36.36; SD = 10.12). The mean length of clinical career was 9.80 years (SD = 7.46).

### Measurements

#### *Patient safety competency*

The patient safety competency scale developed by *Schnall et al. (2008)* and revised and validated by *Lee et al. (2014)* was employed in this study (*e.g.*, "I feel confident in enhancing patient safety through effective communication with other healthcare providers"; "I feel confident in managing inter-professional conflict"). This scale is composed of 14 items which were scored using a Likert scale: 1 = "strongly disagree"; 2 = "disagree"; 3 = "neither agree nor disagree"; 4 = "agree"; and 5 = "strongly agree." Cronbach's alpha coefficient was 0.91 in this study.

#### *The perception of patient safety culture*

The perception of patient safety culture scale, the version modified for Korea by *Lee (2015)*, was employed in this study (*e.g.*, "Clinicians should routinely spend part of their professional time working to improve patient care, Hospital management provides a work climate that promotes patient safety"). This questionnaire is composed of 35 items asking

**Table 1 General characteristics (N = 165).**

| Variable | Category | n (%) |
| --- | --- | --- |
| Gender | Male | 10 (6.1) |
| | Female | 155 (93.9) |
| Participation in the accreditation program for healthcare organizations* | Yes | 40 (24.2) |
| | No | 125 (75.8) |
| Status | Staff | 1 (7) |
| | Nurse | 106 (75.7) |
| | Charge nurse | 18 (12.9%) |
| | Head nurse | 15 (10.7) |
| Department | Ward | 67 (40.9) |
| | Emergency room | 13 (7.9) |
| | Operating room | 25 (15.2) |
| | Ambulatory part | 31 (18.9) |
| | Special part | 28 (17.9) |

| Variable | Means ± SD |
| --- | --- |
| The number of safety education briefings attended | 2.57 ± 2.71 |
| Clinical career | 9.80 ± 7.46 |
| Age | 36.36 ± 0.12 |

nine questions related to leadership, six related to teamwork, five related to patient safety knowledge and attitude, four related to patient safety policy and procedure, four related to non-punishment environment, four related to patient safety improvement system, and three related to patient safety priorities. Items were evaluated with a five-point Likert scale. If the score of an item was high, it means that the awareness level for patient safety was high. The Cronbach's alpha coefficient was 0.93.

### Safety control

The safety control scale developed by *Anderson et al. (2004)* and translated and validated by *Chung (2009)*, to be used for hospital staff, was employed in this study (*e.g.*, "I can take the necessary actions to prevent accidents during nursing"; "I can control myself to follow safe guidelines according to regulations"). This scale is composed of seven items which were scored using a Likert scale: 1 = "strongly disagree"; 2 = "disagree"; 3 = "neither agree nor disagree"; 4 = "agree"; and 5 = "strongly agree.". If the score of an item is high, it means that the safety control is well under control. Cronbach's alpha coefficient was 0.84 in the study.

### Safety education

In this study, safety education means the number of patient safety training sessions received within the last year. In this study, safety education means the number of patient safety training sessions received within the last year. Patient safety training includes education seminars, workshops, and short-term programs to improve the quality and safety of the healthcare systems in hospitals. Hospital nurses must be trained in statutory

duties conducted in the auditorium about once a month, through a small nursing team training, or online training. Face-to-face education is conducted for about 40 people, and these days, most education is being done online due to the coronavirus. The contents of safety education are mainly about communicating effectively, identifying, and managing adverse events and near misses, being ethical, preventing medication errors, infection control, and reporting and sharing mistakes, *etc.*

## Ethical approval

The nurses agreed to join in this study after being informed of the purpose, potential risks, and data collection procedures of the study. We also fully explained in advance that participation could be withdrawn at any time in the interim. The Institutional Review Board approval code of this article is Namseoul Univesity (NSU) 104179-HR-202109-007.

## Data management and analysis

The collected data were analyzed with SPSS 18.0 and AMOS. Several descriptive statistics, including Pearson product-moment correlations, were used. The standard chi-square index of statistical fit, the root means square error of approximation (*Browne & Cudeck, 1993*), the Tucker–Lewis index (*Tucker & Lewis, 1963*), and the Comparative Fit Index (CFI) were utilized to evaluate the fit of structural models to the data. As the patient safety competency scale and safety control scale were one-dimensional scales, item parceling was used to improve the quality of indicators and to reduce the falsifiability of the tested model (*Wu & Wen, 2011*).

## RESULTS

### Descriptive statistics and correlation

Table 2 shows descriptive statistics for the perceptions of patient safety culture, safety control, and patient safety competency of nurses. The number of safety education briefings attended was 2.57 ± 2.71 (min = 0, max = 12). The levels of patient safety culture, safety control, and patient safety competency of nurses were 3.17 ± 0.50, 3.62 ± 0.65, and 4.18 ± 0.52.

Correlations are shown in Table 3. Nurses' perceptions of patient safety culture and safety control were positively associated with patient safety competency. However, the number of times a nurse attended a safety course was not significantly associated with safety control or patient safety competency.

### Model fit

The hypothesized model examined the relationships among nurses' participation in safety education, their perception of patient safety culture, safety control, and their overall patient safety competency. The summarized results of the hypothesized model are shown in Table 4. The results indicate that the model provided a good fit to the data, except for the $\chi^2$ value, which was significant ($\chi^2 = 147.71$, df = 72, $p < 0.001$). While a significant $\chi^2$ test can suggest some level of misfit, this is common in large samples, and other fit indices—such as the Comparative Fit Index (CFI = 0.96), Tucker-Lewis index (TLI = 0.94), and root mean

**Table 2 Descriptive statistics for variables.**

| Variable | Means ± SD |
|---|---|
| The perception of patient safety culture | 3.71 ± 0.50 |
| Safety control | 3.62 ± 0.65 |
| Patient safety competency | 4.18 ± 0.52 |

**Table 3 Correlations among main variables.**

| | Perception of patient safety culture | Safety control | Patient safety competency |
|---|---|---|---|
| The number of safety education briefings attended | 0.20* | 0.10 | 0.14 |
| Perception of patient safety culture | 1 | 0.51** | 0.51*** |
| Safety control | | 1 | 0.41*** |

Notes:
* $p < 0.05$.
** $p < 0.01$.
*** $p < 0.001$.

square error of approximation (RMSEA = 0.07)—demonstrate that the model fits well overall. These values meet the generally accepted thresholds for good model fit, with CFI and TLI close to or above 0.95 and RMSEA below 0.08, indicating a reasonable fit to the data (*Hu & Bentler, 1999*).

## The relationship among nurses' safety education participation, the perception of patient safety culture, safety control, and patient safety competency

The model identified significant correlations between the frequency of safety education and the perception of patient safety culture, as well as significant correlations among perception of patient safety culture, safety control, and patient safety competency (see Table 4). Specifically, nurses who frequently participated in safety education demonstrated a stronger perception of patient safety culture. Moreover, a stronger perception of patient safety culture was significantly correlated with improved safety control and patient safety competency. Similarly, the correlation between safety control and patient safety competency was also significant.

However, the direct relationships between the frequency of safety education and both safety control and patient safety competency were not significant. These findings suggest that while safety education alone does not directly influence safety control or competency, it plays a key role indirectly by shaping nurses' perceptions of safety culture, which in turn influences safety control and competency. Overall, the hypothesized model explained 38% of the variance in nurses' patient safety competency, providing moderate support for the model (see Fig. 2).

**Table 4 Regression weights of hypothesized model.**

| | Estimate (Unstandardized) | Estimate (Standardized) | SE | CR |
|---|---|---|---|---|
| Safety education → the perception of patient safety culture | 0.04 | 0.22** | 0.01 | 3.04 |
| Safety education → safety control | −0.00 | −0.01 | 0.01 | −0.18 |
| The perception of patient safety culture → safety control | 0.73 | 0.63*** | 0.09 | 8.31 |
| The perception of patient safety culture → patient safety competency | 0.21 | 0.21* | 0.08 | 2.39 |
| Safety control → patient safety competency | 0.39 | 0.44*** | 0.08 | 4.77 |
| Safety education → patient safety competency | 0.01 | 0.06 | 0.01 | 1.00 |

Notes:
* $p < 0.05$.
** $p < 0.01$.
*** $p < 0.001$.
SE, Standard error; CR, Composite reliability.

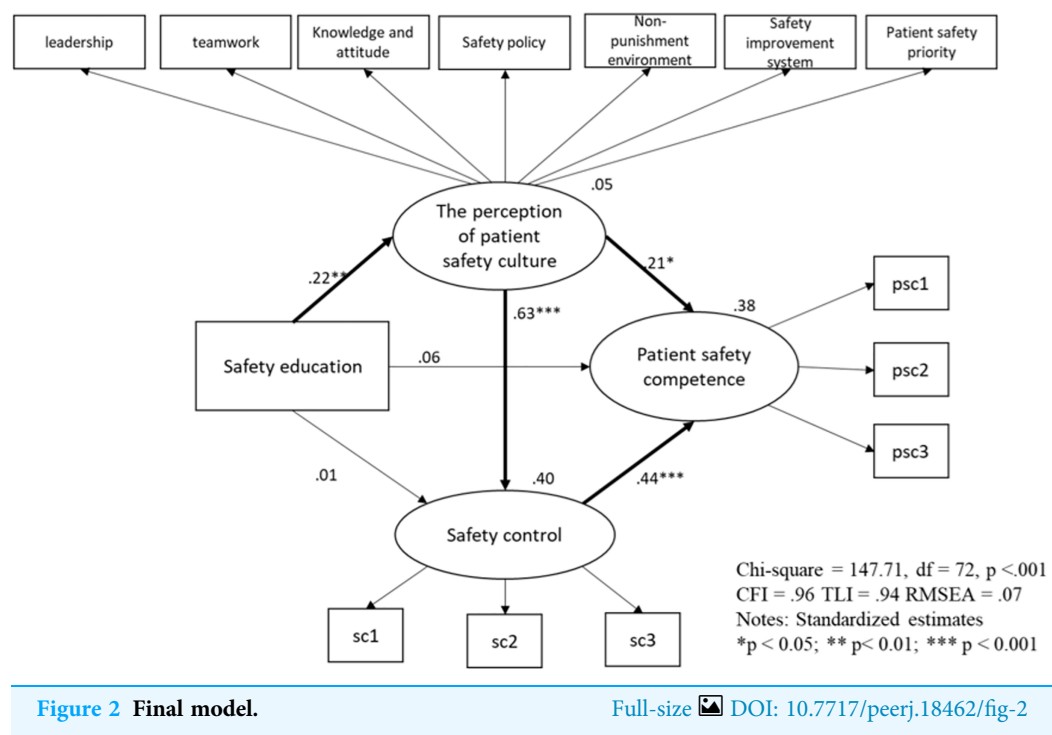

**Figure 2 Final model.**     

## The mediating effect of nurse's perception of patient safety culture and safety control

The indirect effects of the frequency of safety education on patient safety competency were further analyzed, with a focus on the mediating roles of nurses' perception of patient safety culture and safety control (see Table 5). The analysis revealed that frequent participation in safety education indirectly influenced nurses' patient safety competency *via* these mediators. Specifically, nurses who frequently attended safety education sessions were more likely to prioritize patient safety, leading to an enhanced perception of patient safety

**Table 5  Standardized indirect effect.**

| | Indirect effect[+] | Lower bounce | Upper bounce |
|---|---|---|---|
| Safety education → safety control | 0.14 (0.07)** | 0.07 | 0.21 |
| Safety education → patient safety competency | 0.08 (0.04)** | 0.04 | 0.18 |
| the perception of patient safety culture → patient safety competency | 0.16 (0.17)** | 0.17 | 0.40 |

Notes:
  ** $p < 0.01$.
  [+] Unstandardized estimate (Standardized estimate).

culture. This improved perception of safety culture, in turn, positively influenced their safety control, ultimately resulting in higher patient safety competency.

This finding underscores the importance of fostering a positive safety culture and empowering nurses with a sense of safety control, as these factors significantly contribute to their ability to ensure patient safety. Thus, while safety education alone may not have a direct impact, its role in shaping perceptions and safety control is crucial for enhancing nurses' patient safety competency.

# DISCUSSION

This study was conducted to determine whether the number of safety education briefings attended, the perception of patient safety culture, and safety control could predict patient safety competency, and to verify the mediating effects of the perception of patient safety culture and safety control.

## The relationship among nurses' safety education participation, the perception of patient safety culture, safety control, and patient safety competency

First, there was a significant relationship between nurses' safety education participation and the perception of patient safety culture; however, there were no significant relationships among nurses' safety education, safety control, and patient safety competency. This finding was partially consistent with previous studies suggesting the direct relationships among participation in patient safety training, the perception of patient safety culture, and patient safety competency (*Park, Oh & Kim, 2017*). Safety education within the last year was a crucial factor in the perception of the safety culture, patient safety competency, and safety nursing activities (*Kim & Kim, 2017*; *Yan et al., 2021*). However, this study found that the number of safety education briefings attended was not significantly related to safety control ability or safety competency, which could have been due to the content and teaching methods of the safety lessons. Above all, the content of nurse safety education should be composed of what nurses need. Patient safety education should consist of content that helps nurses efficiently cope with patient safety accidents in healthcare. Since most nurses undertake safety education through one-sided lectures or online education in the process of preparing for certification evaluations, it is believed that safety education is somewhat unfamiliar and sometimes difficult, and that education at the institutional level is not enough to effectively improve safety competency (*Kim & Han, 2016*).

In addition, neither the awareness of safe nursing activities nor the safety culture in hospitals is built in a short time. The majority of nurses agreed that professional beliefs in nursing are highly important; however, it was difficult to apply all beliefs in practice (*American Association of Colleges of Nursing, 2024*). The process in which nurses acquire beliefs and practices for patient safety is gradual and continues throughout various experiences in healthcare. From a similar perspective, the reasons for nurses' continued participation in nursing education were to develop their expertise, improve services to patients, and promote solidarity and professional commitment (*Han & Lee, 2010*). Recently, Quality and Safety Education for Nurses (QSEN) emphasized preparing nurses with the competencies necessary to enhance the quality and safety of the healthcare systems (*Cronenwett et al., 2007*). Indeed, it is necessary for hospital policymakers to develop a patient safety training program for nurses from a long-term perspective and to provide nurses with continuing quality education programs.

## The mediating effect of nurse's perception of patient safety culture and safety control

Nurses' patient safety education indirectly influenced patient safety competency *via* a nurse's perception of patient safety culture and safety control, as predicted. However, the correlations between patient safety education and patient safety competency, and patient safety education and safety control, were not significant. This finding is partially consistent with studies on nurses' patient safety competency (*American Association of Colleges of Nursing, 2024*; *Sammer et al., 2010*; *Kim & Han, 2016*; *Kim & Kim, 2017*), which emphasized the importance of promoting the nurses' perception of patient safety culture and safety control abilities to build a safer health system. Additionally, this study proved the mediating effects of a nurse's perception of patient safety culture and safety control in the relationship between a nurse's patient safety education and patient safety competency.

This finding highlights the importance of a nurse's perception of patient safety culture and safety control to improving his/her patient safety competency. A high level of patient safety culture is associated with reducing patient complications and adverse events (*Wang et al., 2014*) and increasing patient safety-related nursing activities (*Park, Kang & Lee, 2012*). Considering that patient safety is a top priority for all members of a medical institution (*Jeong, Seo & Nam, 2006*) and that the most basic principle of patient safety is to form a patient safety culture, nurses need to raise awareness of patient safety culture. Nurses play a critical role in promoting patient safety due to their continuous care for patients and interactions with patients' families and other healthcare professionals (*Patient Safety Network, 2017*). However, the scores of perceptions of patient safety culture in hospital nurses were low in Korea (*Kim, Lee & Choi, 2013*). Recently, various programs have been developed to strengthen the safety culture awareness of nurses, which are effective at reinforcing their role in patient safety culture improvement (*Amiri, Khademian & Nikandish, 2018*). Given that perceptions of patient safety culture are closely related to safety control, organizations need to provide systematic training to improve nurses' awareness of safety culture.

A nurse's safety control is his/her ability to ensure patient safety in healthcare (*Ramanujam, Abrahamson & Anderson, 2023*). The nurses with higher safety control are less likely to put patients in danger; they control themselves appropriately, following the guidelines of safety regulations (*Kim, 2019*). Considering nurses' safety control leads to correct judgments on patient safety (*Kim, 2021*), it is necessary to increase nurses' confidence in safety control. A nurse's safety control is influenced by his/her clinical career. Nurses' work experience and confidence in their tasks are closely related to safety control (*Kim, 2016*). In general, novice nurses lack experience in nursing work and have difficulty coping with unexpected situations. Therefore, a career-specific approach is needed to strengthen a nurse's safety control.

A nurse's perceptions of patient safety culture and safety control form an individual factor. No matter how good and often an institution provides safety education, the effectiveness of said education at promoting nurses' safety competency can vary depending on how an individual accepts it and internalizes it. Nurses' participation in safety education did not directly affect their patient safety competency; however, the nurses' safety education influenced their awareness of patient safety culture. Even though individual differences exist, participation in safety education and activities should improve nurses' awareness of safety culture-related activities, such as open communication and teamwork, and a positive perception of safety culture could reinforce safety control, patient safety capabilities, and finally, safe nursing activities. A change in perception leads to a change in behavior. When organizing programs to enhance nurses' safety culture competency, contents, and teaching methods are specified for novice and experienced nurses. When designing educational programs, it is necessary to focus on recognizing the importance of a safety culture for novice nurses and to address various cases that occur in real-life nursing situations for experienced nurses. Considering that nurses' patient safety capabilities result in a safer healthcare system, specific and effective educational interventions should be provided to improve nurses' perceptions of patient safety culture and strengthen safety control.

## Limitation

This study focused on the variables of personal characteristics that influence the nurse's patient safety capabilities. However, since nurses' patient safety capabilities are also affected by environmental factors such as national medical policies and hospital medical policies, future studies need to examine variables affecting nurses' patient safety capabilities from an ecological perspective (microscopic, intermediate, and macroscopic perspective). Also, there is a limitation in that the nurse's safety education, which was a major variable in this study, could not take into account the qualitative differences in education as various types of education (*e.g.*, workshops, seminars, online lectures, and so on) were included. In future studies, it is necessary to examine whether there are differences in patient safety competency according to the type of education and teaching method.

## CONCLUSIONS

In this study, we explored the direct and indirect relationships among variables influencing the patient safety competency of nurses in Korea. Our findings emphasize the critical role of nurses' perceptions of patient safety culture and their sense of safety control in enhancing their overall competency in patient safety. These insights can serve as a foundation for encouraging nurses to improve their safety competencies through targeted educational and organizational interventions.

However, it is important to note that this study did not account for nurses' actual safety-related behaviors or activities in clinical practice. Future research should consider incorporating environmental factors, such as institutional policies and organizational support systems, which may significantly impact patient safety outcomes. By addressing these aspects, future studies can provide a more comprehensive understanding of how to strengthen patient safety in healthcare settings.

## IMPLICATIONS

First, given the significant influence of patient safety culture on nurses' safety competency, healthcare institutions should prioritize fostering a positive safety culture. Leaders should actively promote safety-centered behaviors and create an open communication environment where nurses feel comfortable sharing their concerns and reporting incidents without fear of retribution. Second, training programs should focus on giving nurses the authority and resources to take proactive roles in addressing safety concerns in clinical practice. In addition to safety education, institutional policies that support patient safety must be considered. Healthcare organizations should evaluate the impact of their policies on nurses' safety-related activities and provide adequate support and resources to enhance patient safety practices.

## ACKNOWLEDGEMENTS

This is a short text to acknowledge the contributions of specific colleagues, institutions, or agencies that aided the efforts of the authors.

### Funding

This research was supported financially by Namseoul University. The funders had no role in study design, data collection and analysis, decision to publish, or preparation of the manuscript.

### Grant Disclosures

The following grant information was disclosed by the authors:
Namseoul University.

### Competing Interests

The authors declare that they have no competing interests.

## Author Contributions

- Jung-hyun Choi conceived and designed the experiments, performed the experiments, analyzed the data, prepared figures and/or tables, authored or reviewed drafts of the article, and approved the final draft.
- KyoungEun Kim conceived and designed the experiments, performed the experiments, analyzed the data, prepared figures and/or tables, authored or reviewed drafts of the article, and approved the final draft.

## Human Ethics

The following information was supplied relating to ethical approvals (*i.e.*, approving body and any reference numbers):

Informed consent was obtained from all subjects involved in the study. The Institutional Review Board approval code of this article is NSU 104179-HR-202109-007.

## Data Availability

The raw measurements are available in the Supplemental File.

## Supplemental Information

Supplemental information for this article can be found online at http://dx.doi.org/10.7717/peerj.18462#supplemental-information.

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
