# Peer review of "A structural equation model for the patient safety competency of clinical nurses"

_PeerJ, doi:10.7717/peerj.18462_

## Round 0.1 · original submission · Major Revisions

Your paper is very interesting and important research but before we can consider further for publication, you will need to address the following:
* Please address the comments from all reviewers 1 and 3, specifically, shorten the introduction and briefly describe the rationale
* Please describe the sample size estimation, on what basis did you decide to include 191 nurses
* In the conceptual model of the safety, please clearly indicate which ones are latent variables and which are the manifest variables
* In the structural model, label each path with the path coefficients
* Please comment on the identifiability and specification of the model. You may refer to the following journal article on how to report
https://www.ncbi.nlm.nih.gov/pmc/articles/PMC3755633/
* You also need to present the fit indices (the reviewers did not ask for this but it is important to write these details for the benefit of the readers to evaluate your findings)
* The results and discussion section needs to address the findings from your models

·

Basic reporting

In abstract section, line 32, add KEYWORDS.

Experimental design

In introduction section, line 150, add research questions.
In Materials & Methods section, add description of the general characteristic of the studied nurses (gender, Participation in The Accreditation Program for Healthcare Organizations, Status, and Department) to Measurements.
In Materials & Methods section, line 156, add duration of data collection to Sampling and Data Collection.
In Materials & Methods section, line 160, more details about study sample like sample size, method of selection, if there any inclusion or exclusion criteria should be included.

Validity of the findings

No comment

Additional comments

In discuss section, line 276, the citation of reference should be (park, oh, & kim, 2017)
In reference section, line 472, revised this reference.

Reviewer 2 ·

Basic reporting

No comment.

Experimental design

no comment

Validity of the findings

No comment

Additional comments

Dear authors,

the research study presents variables in relation to patient safety where nurses have important competence. The study highlighted nurses' management of a patient safety culture, establishing the need for education and ensuring an environment and control for patient safety.
Adequate measurement tools used and statistical evaluation of the results testify to the mastered rigour of the research.

·

Basic reporting

Dear authors,
Congratulations on conducting a very important study.
Kindly refer to the attached file for improvements to this study.
All the best!

Experimental design

"Not relevant to this study"

Validity of the findings

"no comment"

Additional comments

Please refer to the attached file.

Reviewer 4 ·

Basic reporting

The English in your text is generally clear and professional. The literature cited provides a thorough understanding of patient safety issues, competency development, and organizational culture.

Experimental design

the paper reviewed is a descriptive correlational study report.
The research methods described are sufficient.

Validity of the findings

The results presented are generally valid, with a good fit for the hypothesized model and meaningful relationships among the variables.
The direct impact of safety education on safety control and competency could be further explored, possibly through improved measurement or more focused educational interventions.

Additional comments

The study provides valuable insights into the indirect effects of safety education on patient safety competencies through perceptions of safety culture and control.
addressing the limitations and expanding the scope of research could enhance the understanding and effectiveness of patient safety interventions.
Emphasizing the role of nurses’ perceptions of safety culture and control in the effectiveness of safety education provides valuable insights for improving training programs.

---

## Round 0.2 · accepted · Accept

I can confirm as per reviewers that the authors have addressed all of the reviewers' comments, and I have reviewed the manuscript for amendments as well. The manuscript is ready for publication.

·

Basic reporting

Greetings and Regards
Amendments are approved

Experimental design

No comment

Validity of the findings

No comment

Additional comments

No comment

·

Basic reporting

The authors have addressed the given suggestions and improved the manuscript. Therefore, there are no more comments for this study.

Experimental design

No comments

Validity of the findings

Authors have addressed the validity findings.

Additional comments

This is a very good study and looking forward to seeing this study publication.